# Modeling Study of a Microbial Spray-Drying Process Based on Real-Time Sampling

Feng-kui Xiong [1], Yue-jin Yuan [1,*], Ying-ying Xu [1], Jing-yu Li [2], Zhe Zhao [1] and Li-bin Tan [1]

1   College of Mechanical & Electrical Engineering, Shaanxi University of Science & Technology, Xi'an 710016, China
2   College of Land and Environment, Shenyang Agricultural University, Shenyang 110866, China
*   Correspondence: yyjyuan1@163.com; Tel.: +86-15-8290-45176

**Abstract:** The process of microbial spray-drying has inherent defects such as short time, complexity, and non-visualization of particle trajectory. However, there has been a lack of effective methods for real-time sampling, rehydration, and non-destructive storage of dried particles, as well as mathematical modeling of the drying process of yeast particles based on sampling and measurement data. Therefore, firstly, a real-time sampling system was developed which completed real-time sampling, rehydration, and non-destructive storage of spray-dried particles, and realized the real reproduction of the changes of yeast particles in the process of spray drying. The laws that the motion trajectory of microbial particles during spray drying are divided into the first cycle region and the reflux cycle region were concluded, and the partition mechanism was explored. Then, based on the sampling data and the law of heat and mass transfer, a mathematical model of porous media was established to predict the variation of moisture content and rehydration survival rate of dried microbial particles with drying time. Finally, the mathematical model was tested by a microbial spray drying experiment, and the maximum errors between the predicted value and the test value of moisture content and rehydration survival rate were $X_{\max1} = 0.027(d.b)$ and $Q_{v\max1} = 1.06\%$, respectively, both were less than 5% which proved the correctness of the mathematical model of porous media and laid a foundation for the study of the damage mechanism of microbial spray drying.

**Keywords:** microbial spray-drying; real-time sampling; laws of heat and mass transfer; porous media mathematical prediction model





## 1. Introduction

Microbial desiccation is the core research topic of livelihood fields (such as food, medicine, and etc.). As the most widely used microbial drying process, spray drying has the advantages of high efficiency, low cost, and continuous production. It not only occupies a dominant position in the market, but also is a hot spot in current research [1]. Nevertheless, due to the inherent defects of the microbial spray drying process, such as short time, complexity, and invisible trajectory of spray particles [2], it is very difficult to directly sample and study the drying process of microbial particles. Therefore, there has been a lack of effective means for real-time sampling, rehydration, and non-destructive storage of dried particles, and further establishing mathematical models to accurately predict the moisture content and rehydration survival rate of microbial particles in the spray drying process [3]. At present, the common research method is "single droplet simulation experiment + mathematical modeling" [4]. On the one hand, the single droplet simulation experiment could not reproduce environmental components such as short residence time, complex movement trajectory, and high-speed collision of spray-dried particles, and the experimental data could not truly reflect the changes of microorganisms in the spray-drying process. On the other hand, most mathematical models ignore changes

in the trajectory of a dry particle and simplify the trajectory to top-down homogeneous motion [5]. Therefore, the mathematical models established by this combination method could not predict the spray-drying process of microorganisms well.

Dr. Gong Pi-min [6], (Harbin Institute of Technology), proposed a vertical arrangement of a single-cup straight cylinder sampling system. Although this system could complete the real-time sampling of spray-dried particles, it suffered from the disadvantages of poor sampling stability and nondestructive storage, which could not reproduce the real microbial changes during spray-drying. Previous research on modeling has mainly focused on regression analysis to build empirical models, such as Akpinar [7], equal to the thin layer model proposed in 2006, there was little in-depth exploration of heat and mass transfer and microbial cell damage of the drying process, so the prediction accuracy was not high [8]. In 2013, L. Spreutels [9], a Canadian scholar, established an image-only mathematical model [10], based on the dynamic drying curve method based on the single droplet simulation experiment, although the spray particles were assumed to fall at a uniform speed in the spray drying process. However, the model was based on the single drop test, so the rehydration survival of yeast particles during drying could not be accurately predicted as a function of drying time [11].

In response to the above problems, firstly a novel microbial spray-drying real-time sampling system (submitted for a Chinese invention patent) was developed which could implement real-time sampling, rehydration, and nondestructive storage on spray-dried particles. Then, on the basis of real-time sampling data, the heat and mass transfer model of porous media proposed by the research team in the previous study was used to complete the mathematical modeling of the spray-drying process. Finally, through the comparison with and the analysis of the experimental data, it was proved that the model could accurately predict the moisture content and rehydration survival rate of dried particles with time in the process of microbial spray drying.

## 2. A Brief Introduction of the New Sampling System

### 2.1. A Brief Introduction of the Sampling Arrangement of Double Wet and Dry Samples in Step-Ladder Type

In the novel sampling scheme, a dry-pellet sampling cup and a rehydration sample sampling cup, 50 mm apart, were arranged on the same ladder step and were responsible for collecting the dry pellets and real-time rehydration of the dry pellets, respectively, with nondestructive storage. The drying tower was then bisected equally by height and perimeter into 6 elevations and 7-segment arcs, respectively, with each ladder step again arranged at the intermediate value of height and radians, completing the 7-stage sampling arrangement. In order to prevent the spray particles from directly entering the sampling cup and affecting the sampling stability, arc-shaped baffles were installed at 30 mm above the mouth of each sampling cup. Finally, the design of the stepped-type dry and wet double-sample sampling arrangement was completed. This sampling scheme, with a step ladder-type design, avoids interference between the individual sampling cups in a vertical-type arrangement scheme, which, combined with the installation of arcuated baffles, greatly improves the stability and accuracy of sampling.

### 2.2. A Brief Review of Two Cup Surface Sampling Cups

In terms of sampling principle, the sampling cup attracts dry particles loaded with microorganisms and carboxyl iron powder (CIP) into the sampling cup through the magnetic field of a strong magnet at the bottom of the outer cup, so as to achieve the purpose of stable sampling [12].

At constant temperature and humidity, the sampling cup was filled with $-40\,^\circ$C antifreeze for mobility in the freezing layer, and then put into the $-60\,^\circ$C refrigerator for quick freezing to form a solid state, so as to maintain the low-temperature environment in the inner cup. At the same time, the thermal and magnetic exchange between inside and outside of the sampling cup was isolated by pasting the thermal insulation magnetic

patch evenly on the outer surface of the outer cup, and the low-temperature environment in the inner cup was further maintained, while the strong magnetic field was prevented from penetrating the outer cup and affecting the sampling process.

In order to improve the sampling stability and the capacity of the water to hold a constant temperature at the same time, the sampling cup was improved to design a straight cylinder inner cup to be a convex curved inner cup by rotating the first 20 mm of the Bezier cubic curve [13], with control points $P_0 = (0, 0)$, $P_1 = (12, 9)$, $P_2 = (24, -16)$ and $P_3 = (48, 8)$. The curve group is shown in Figure 1 below, and the equation of curve group is shown in Equations (1)–(3) below [14]:

$$P(\mu) = \sum_{i=0}^{n} B_i^n(\mu) P_i \tag{1}$$

$$B_i^n(\mu) = \frac{n!}{i!(n-1)!} \mu^i (1-\mu)^{n-i} \tag{2}$$

$$P(\mu) = \mu^0 (1-\mu)^3 P_0 + 3\mu(1-\mu)^2 P_1 + 3\mu^2(1-\mu) P_2 + \mu^3(1-\mu)^0 P_3 = (1-\mu)^3 P_0 + 3\mu(1-\mu)^2 P_1 + 3\mu^2(1-\mu) P_2 + \mu^3 P_3 \tag{3}$$

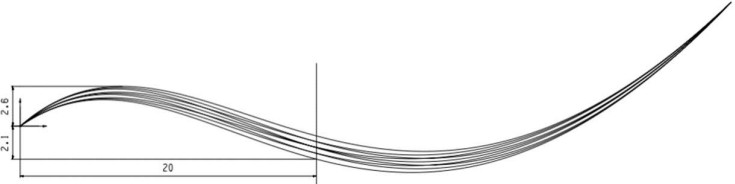

**Figure 1.** Schematic diagram of Bezier cubic curve.

In the Equations (1)–(3): $n$—degree of polynomial; $i$—non-negative integer; $P_i$—control point representing curve equation; and $B_i^n(\mu)$—The basis function of the nth order curve equation.

The curved inner cup structure changed the air flow field, wind speed, and pressure field in the cup and reduced the inflow of external hot air (as shown in Figure 2). Finally, the purpose of improving the constant temperature water holding capacity of the sampling cup without increasing the depth of the sampling cup to ensure the sampling capacity and stability was achieved.

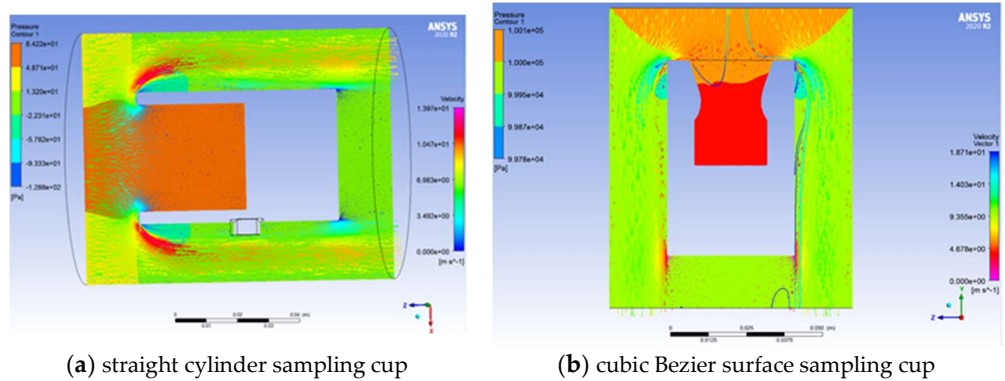

(**a**) straight cylinder sampling cup    (**b**) cubic Bezier surface sampling cup

**Figure 2.** Comparison of pressure field and wind speed field between straight cylinder sampling cup and curved sampling cup.

In order to achieve non-destructive installation, the sampling cup was fixed by connecting a magnet and spray-drying tower to achieve the purpose of non-destructive installation. In addition, the four exposed surfaces of the magnet were affixed with thermal insulation magnetic patches to prevent them from affecting the spray-drying sampling process.

In order to measure temperature and humidity in real time, the sampling cup recorded the sampling temperature and humidity at the sampling point in real time through a pasted

wireless temperature and humidity sensor placed in the outer cup mouth. The curve of temperature and humidity change was drawn.

The overall structure diagram of the sampling cup is shown in Figure 3 below. The dimensions of the sampling cup are: the upper surface of the inner cup is 20 mm high, the minimum inner diameter is 18 mm; the lower part of the inner cup is 10 mm high and the inner diameter is 24 mm; the height of the outer cup is 65 mm, the outer diameter is 42 mm; and the wall thickness is 2 mm.

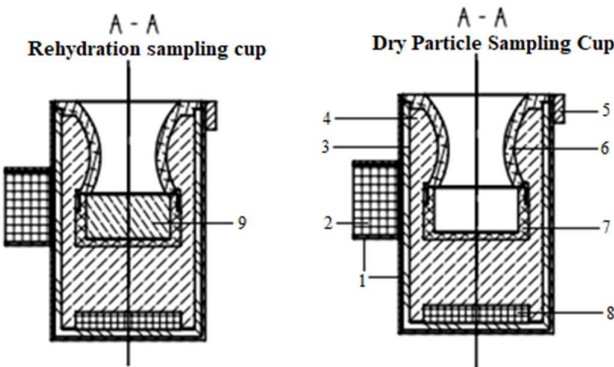

**Figure 3.** Diagram of a new double surface sampling cup structure. (1) Heat and magnetic insulation tape; (2) Connecting magnet; (3) Outer cup; (4) Frozen layer; (5) Wireless temperature and humidity sensors; (6) Upper part of inner cup; (7) Lower part of inner cup; (8) Powerful magnets; (9) Saline.

A large number of experiments have proven that the combination of the double cup curved sampling cup and step-ladder type sampling scheme could achieve stable real-time sampling, rehydration, and non-destructive storage of spray-drying particles, and the sampling data could accurately reflect the changes of microorganisms in the process of spray drying.

## 3. Experimental Design and Data Analysis

In this test, Baijiu yeast was sampled by a novel sampling system using three parameters: spray-dried material flow, hot air wind speed, and temperature as test variables, and the temperature and humidity at each sampling site were recorded. Then, the dry-based water content, moisture status and particle size of the dried pellets in each dried pellet sampling cup were determined, respectively, and the rehydration survival rate of yeasts in each rehydrated sample sampling cup were determined. Finally, the determination parameters were analyzed to obtain the movement law of microbial particles during spray drying.

### 3.1. Materials and Methods

The schematic diagram of the spray-drying test device is shown in Figure 4. The left figure is the overall structure diagram, and the right figure is the structure diagram of sampling and temperature and humidity detection in the drying tower. The parameters of the spray-drying tower used in the experiment are tower height $h$ = 900 mm, cylinder diameter d = 400 mm, conical taper at the bottom $a$ = 60°, and tower bottom diameter d = 150 mm.

As shown in Figure 4 above, the temperature and humidity measurements of the sampling point were completed by the temperature and humidity sensors arranged at the mouth of the sampling cup in cooperation with the data acquisition system. The dry base moisture content, water state, particle size, and rehydration survival rate of samples in the sampling cup were determined by water meter, differential scanning calorimeter (DSC), Winner3001 dry powder laser particle size meter, and hemocytometry, respectively.

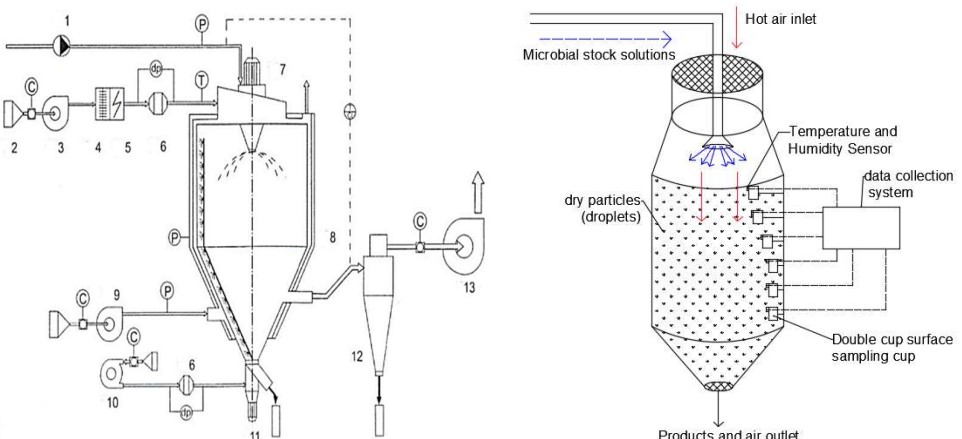

**Figure 4.** Schematic of the parallel flow drying column structure. (1) Feed pump, (2) & (6) Filter, (3) & (9) Fan feeders, (4) Steam heaters, (5) Electric heaters, (7) Nebulizers, (8) Drying column, (10) Air pump, (11) Gas sweep device, (12) Cyclone separator.

### 3.2. Test Procedure

The purpose of this test is to preliminarily analyze the changes of microorganisms represented by Baijiu Yeast in the drying process by measuring the values of various evaluation indexes of samples in different sampling cups under various test variables, so as to lay a foundation for the establishment of a mathematical model for predicting the drying process [15].

### 3.2.1. Preparation of Bacterial Solution to Be Dried

According to the research team's prior studies: ① The optimized N medium formulation were yeast extract 15 g/L, peptone 15 g/L, beef extract 15 g/L, fructose 40 g/L, KH2PO4 3 g/L, and inositol 0.1 g/L. ② Yeast desiccant protectant formulations were sucrose 100 g/L, trehalose 100 g/L, lactose 120 g/L, and sorbitol 80 g/L. ③ The optimal vehicles when dried were reconstituted skimmed milk powder (RSM) at a concentration of 30% in culture broth [16]. ④ Carboxyiron powder (CIP) was added as follows: CIP: RSM = 1:8 [17].

The steps to formulate the broth to be dried are: ① Baijiu Yeast was inoculated in 12 groups of 200 mL of novel n medium at an inoculation ratio of 0.5%, then placed into a constant temperature shaking incubator at 28 °C and incubated at a rotation rate of 180 r/min for 9 h. ② Aliquots were centrifuged for 10 min in a centrifuge at 4000 r/min to remove supernatant, remove bacteria, and wash twice by centrifugation in normal saline. ③ The collected bacteria were re-suspended in 20 mL PBS buffer. ④ To the cell suspension, 180 mL of culture solution containing 30% RSM was added, and then CIP was added at a CIP: RSM ratio of 1:8 to mix well. ⑤ According to the above formula of protective agents, protective agents were added, and then placed in a 4 °C freezer for refrigeration.

### 3.2.2. Spray-Drying Test Design

The experimental procedures for spray-drying were: ① Before each test, the components of the spray-drying system were cleaned in full accordance with the cleaning requirements of the spray-drying system. Then the spray-drying system was preheated by predetermined test inlet hot air temperature and flow rate for 10 min to obtain stable test conditions. ② Set the feed pump parameters by the predetermined mass flow and then start the feed pump, starting the test. ③ The temperature and humidity values of each sampling point displayed by the collection system were recorded. After the test was completed, the samples in the sampling cup were immediately taken out and placed in the refrigerator at 4 °C after numbering. ④ The dry-basis water content, moisture status, particle size, and rehydration survival of each sample were determined separately.

First, the yeast particles in the sample cup were put into the crucible of the moisture content measuring machine and dried at 120 °C for 7 h. Then, the mass $m_3$ and $m_4$ of Baijiu yeast granules before and after drying were read out, and then the CPI mass was deducted proportionally and recorded. Finally, calculation of the dry-basis moisture content of the sample particles was performed according to the following Equation (4):

$$\text{Moisture content } (\%) = \frac{m_3 - m_4}{m_4} \times 100\% \qquad (4)$$

The method for measuring the particle size of dry particles is as follows: first, the sampling hopper was inserted into the sampling port of the dry powder sampler of winner 3001 dry powder laser particle sizer. Then, the dry particles mixed in the feed tank were filled up through the feed hopper, and after switching to 1.0–300 µm, the operation button was clicked. Finally, the particle size distribution and average value of dried yeast particles were automatically measured by the laser particle size analyzer, and a report was automatically generated. The average particle diameter of the dried particle sample in the first sampling cup was the input parameter—initial diameter of yeast particles ($D_0$).

The measurement method of moisture state is as follows: first, the scanning temperature of DSC was set to 0–160 °C, and the temperature rise rate is 5 °C/min. Then, 0.1 g of dry particle sample was put into an aluminum crucible for sealing and then was placed into a DSC scanner for scanning. Finally, with the blank aluminum crucible as the blank control, the bound water content and free water content of the dried particles were automatically determined. The content value of the combined water in the water state, i.e., the water content at the transfer point, was taken as the input parameter.

The moisture content, moisture state, and particle size of the dried particles could be determined directly by a moisture meter, DSC, and laser granulometry, respectively, according to standard methods. The rehydration survival rate ($Q_v$) could be calculated according to the following Equation (3) after counting the number of colorless bacteria ($N_{us}$) and the total number of bacteria ($N_T$) in the four top corner counting areas of the blood cell counting board by methylene blue staining + blood cell counting method [18]. The equation is as follows:

$$Q_v = \left\langle \frac{N_{us}}{N_T} \right\rangle \bigg|_{three \cdot fields} \times 100\% \qquad (5)$$

The 16 groups of experimental parameters formed by the combination of 3 test variables and their representative patterns when mapping is shown in Table 1 below (The intersection diagram of the three groups of data in Table 1 is a representative icon of the results obtained after the experiment of the experimental parameters composed of the three groups of data in the subsequent mapping). Three repetitions shall be made for each group of experiments. a total of 48 groups of spray-drying experiments shall be completed, and 1392 test data shall be measured and recorded.

**Table 1.** Test variable group setting and representative pattern.

| Inlet Hot Air Temperature (°C) | | 95 | | 105 | | 115 | | 125 | |
|---|---|---|---|---|---|---|---|---|---|
| Hot air inflow (m³/h) | 60 | □ | ■ | △ | ▲ | ◇ | ◆ | I | + |
| | 70 | ○ | ● | ▽ | ▼ | ☆ | ★ | * | × |
| Material inflow (mL/h) | | 600 | 700 | 600 | 700 | 600 | 700 | 600 | 700 |

### 3.3. Test Data Analysis

The spray-drying experiment of Baijiu Yeast according to the above experimental design was carried out, and the temperature at the sampling point was recorded. Then, the moisture content, particle size, and the survival rate of yeast in rehydrated samples were determined respectively. Finally, the analysis with Origin is shown in Figures 5–8.

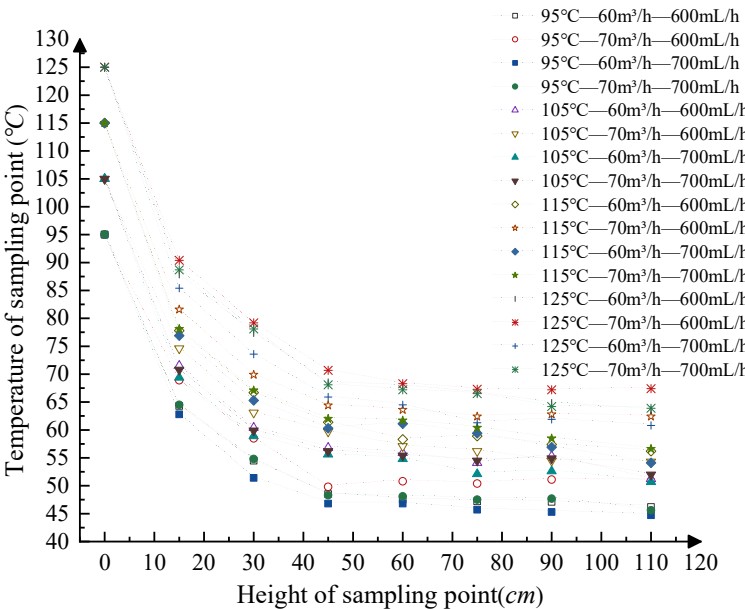

**Figure 5.** Sampling point temperature of each sampling cup under different test parameters.

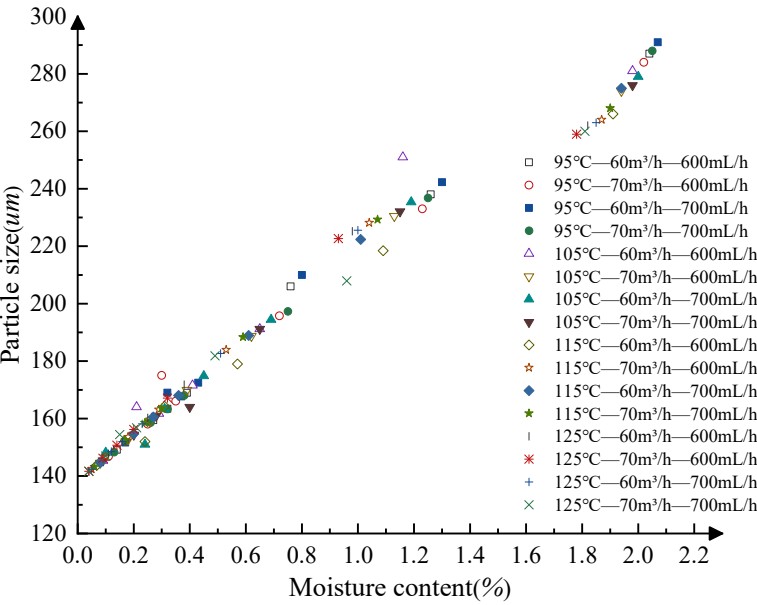

**Figure 6.** Variation of yeast particle size with moisture content.

As shown in Figure 5 above, with increasing sampling height, the measured sampling point temperatures at the cup mouth of the sampling cup all showed a decreasing trend of speed reduction. The rate of temperature drops at the sampling site decreased obviously after sampling height of 15 cm and stabilized near the outlet temperature at sampling site temperature when the sampling height was 45 cm.

As shown in Figure 7, when the sampling height was less than 45 cm, the dry-based water content of yeasts all decreased rapidly with increasing sampling height. After sampling height was greater than 45 cm, under each drying parameter, with the increase of sampling height, the dry-basis water content of yeast, however, showed different variation rules as follows. ① In curve 2 of group-c and all curves of group-d, the dry-basis moisture content of yeast decreased with the increase of sampling height. ② In all the curves except curve 2. In group-b and group-c, the moisture content of dry basis at the sampling height of 75 cm is lower than that at 90 cm. ③ In all curves of group-a, the moisture content of dry basis is lower than 75 cm at the sampling height of 60 cm and then lower than 90 cm.

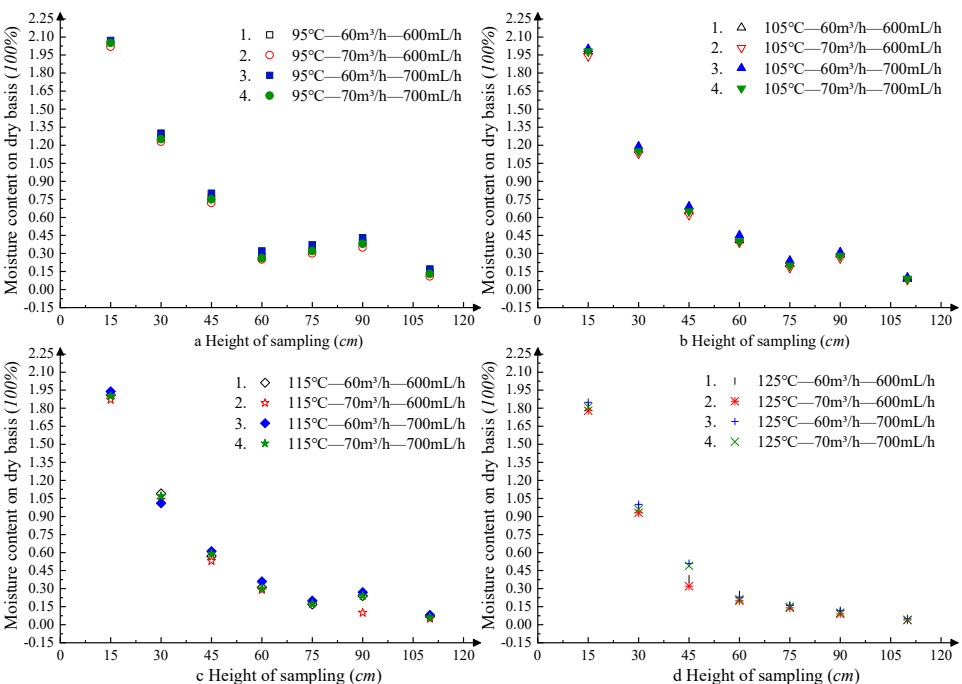

**Figure 7.** Moisture of yeast in each sampling cup under different test parameters.

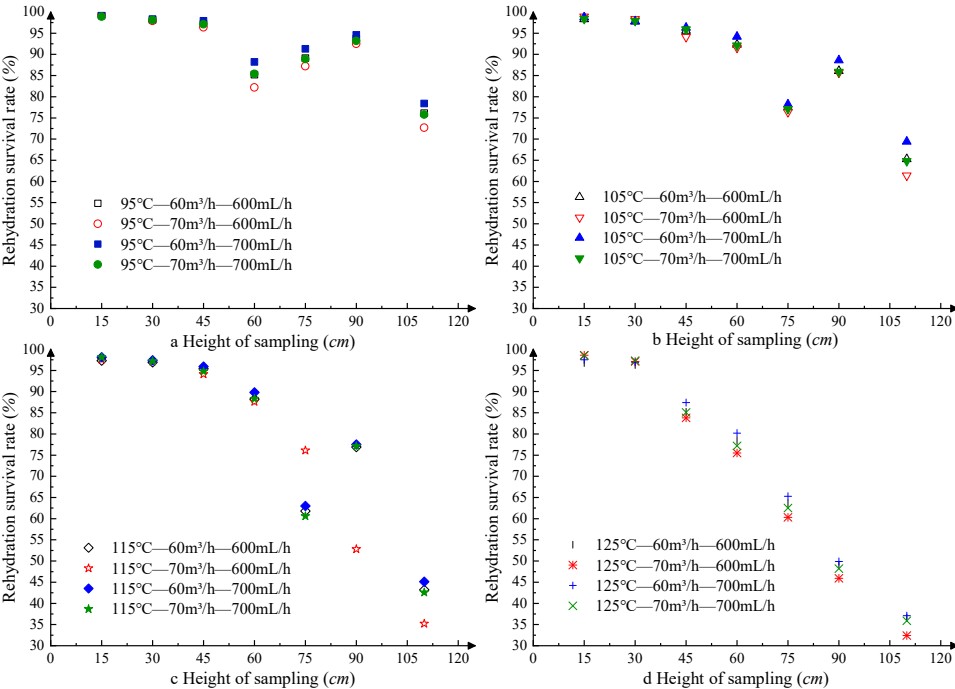

**Figure 8.** Rehydration survival of yeast in each sampling cup under different test parameters.

Based on the analysis of Figures 5 and 7, it could be concluded that in the constant rate drying stage, yeast particles were rapidly dehydrated in a high-temperature environment, which makes the temperature in this stage decrease rapidly, and the sampling height was 45 cm, which is close to the outlet temperature of spray-drying. In the subsequent stage, yeast particles were dried at a low rate in a medium- and low- temperature environment close to the outlet temperature.

It was assumed that the time for the droplet to move from one sampling point to another is equal, which is 2 s. If quadratic interpolation was performed at the time point

without data, the change law of moisture content and rehydration survival rate of yeast dry particles with time could be obtained, as shown in Figures 9 and 10.

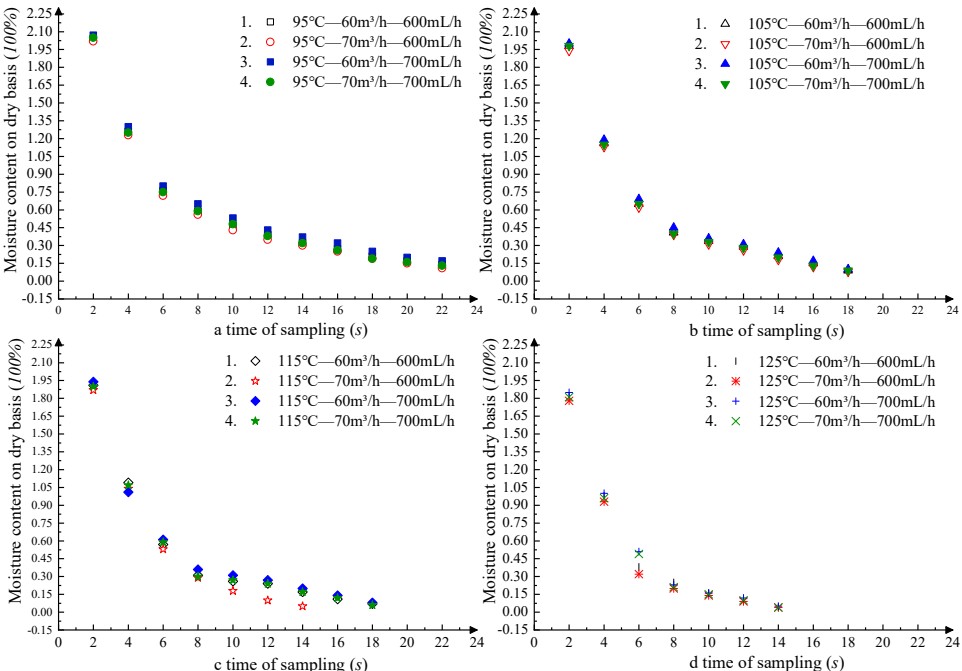

**Figure 9.** Moisture content of yeast under different test parameters with time.

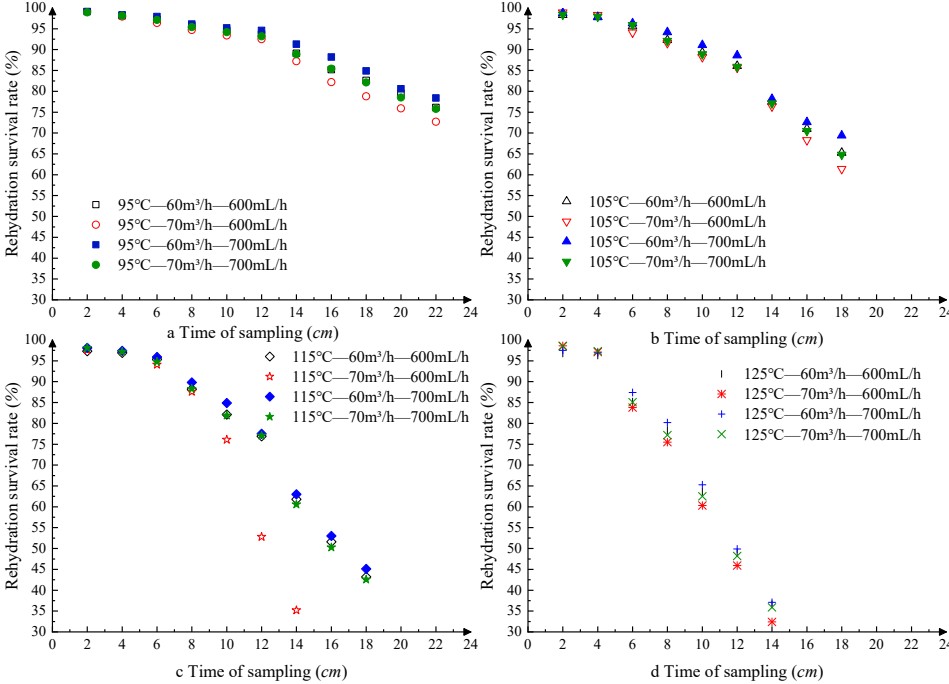

**Figure 10.** Rehydration survival rate of yeast under different test parameters with time.

As shown in Figure 9 above, the moisture content of dry particles decreases with the increase of drying time at a constant rate first and then at a reduced rate.

Combined with the analysis of Figures 7 and 9, yeast particles was not entirely top-down moving. Firstly, in group-c curve 2 and all curves in group-d, yeast particles move from top to bottom without reflux. Secondly, in all the curves except curve 2 in group-b and group-c, yeast particles refluxed from the sampling height of 90 cm to 75 cm, and then

decreased along the tower wall. Finally, in all curves of group-a, the yeast particles refluxed from the sampling height of 90 cm to 60 cm, and then decreased along the tower wall. The motion track is shown in Figure 11.

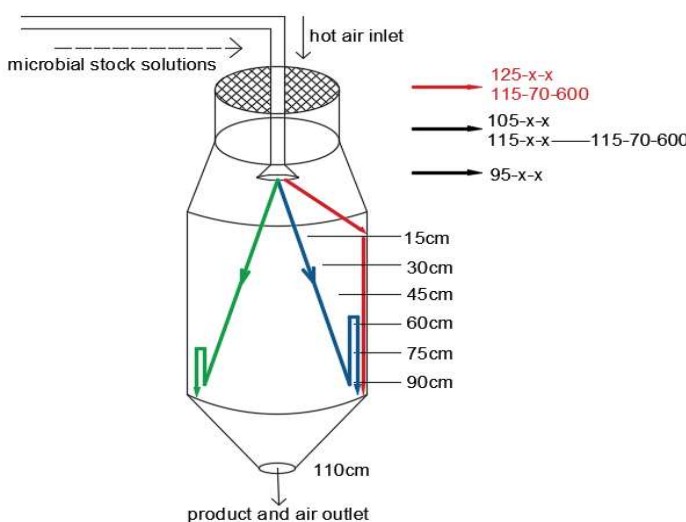

**Figure 11.** Schematic diagram of yeast particle trajectory under three groups of parameters.

By comparing Figures 8 and 10, it could be seen that in the high-speed and constant-speed drying stage, the surface moisture of bacteria particles was sufficient to achieve gas-liquid two-phase equilibrium, and the contact temperature of yeast is the wet bulb temperature, lower than the hot air temperature. At this stage, the main cause of cell inactivation was dehydration damage caused by rapid dehydration. The survival rate of yeast is high and was less affected by drying process parameters. In the stage of reduced rate drying, the moisture on the particle surface could not meet the requirements of gas–liquid two-phase equilibrium, and the moisture inside the particle then migrates to the surface through the pore structure. With the increase of migration resistance, the particle drying rate also decreases. At this stage, when the bacteria were exposed to hot air, the main causes of cell inactivation are heat damage and dehydration damage. The survival rate of yeast was greatly affected by the outlet temperature.

## 4. Mathematical Modeling of the Spray-Drying Process

Based on the test data measured by the new sampling system after sampling, the spray-drying process of Baijiu yeast was modelled to establish a mathematical model that could accurately predict the changes of particle size, water content and rehydration survival rate of Baijiu Yeast. Finally, it was generalized to apply the prediction of all-microbial drying, which is the core purpose of this study.

### 4.1. Common Data Values and Assumptions

To establish the mathematical prediction model based on the experimental parameters, reasonable assumptions must be made in combination with the experimental data, and then the physical model of the spray-drying process and yeast particles should be simplified. Therefore, based on the experimental results, five hypotheses were made as follows.

(1) According to the analysis of water content and rehydration survival rate of yeast particles, the water content in yeast particles could be divided into free water (Class E), intracellular bound water (Class D), and residual water (Class r), as shown in Figure 12 below. The figure on the right is the schematic diagram of yeast spray-drying particles, and the figure on the left is the abstract model diagram of drying particles.

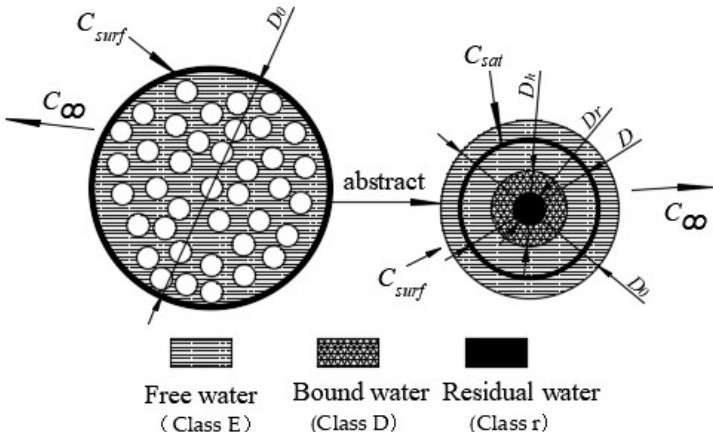

**Figure 12.** Schematic diagram of Baijiu yeast granule drying model. $C_\infty$. Water vapor concentration in the air far from the pellet, $C_{surf}$. Water vapor concentration in the air in contact with the external pellet surface, $C_{sat}$. Saturation concentration of water in pellet, $D$. Real time diameter of yeast particles, $D_h$. Diameter of bound water, $D_r$. Diameter of the residual water, $D_0$. Yeast particle initial diameter.

Since cell membrane is the most typical semi-permeable membrane, which has no effect on the entry and exit of small molecules such as water molecules, in this study, the water in the outer protective agent of cell membrane and the flowing water that cells do not combine with cell structures were uniformly classified as Class E free water [19]. According to the pore network theory, it was assumed that the pore is uniformly distributed in a microscopic pore network (pore throat) with a diameter of 0.1 µm, and its loss was mainly obstructed by capillary forces in porous media and saturated vapor pressure on the particle surface. The presence of Class E water maintained the gas–liquid two-phase balance on the surface of dried particles, and kept the bacteria in the "wet bulb temperature" at the early stage of drying. The yeast was almost not damaged by heat (survival rate > 95%).

Class D water is the bound water in the bacterial body (excluding Class r bound water remaining after drying), which binds to each cellular substructure through hydrogen bounding or intermolecular forces, etc., and acts to maintain the normal function of each cellular substructure. Its dispersion and loss were mainly impeded by the adsorption of cellular substructures, capillary forces in the porous medium and saturated vapor pressure at the particle surface. The loss of Class D water leads to damage to cellular substructures, so the loss of Class D water is the main cause of dry inactivation of yeasts. It was assumed that the two types of water are lost in the order of E and D, and the remaining water (Class r) after drying is all Class D water. Because $t_{md}$ and $t_c$ are much smaller than $t_{dr}$, it was considered that the mass transfer inside and outside the particles is in a quasi-steady state.

(2) Since the radius of the grid channel of the protective agent outside the yeast body is 10 µm, the capillary effect has negligible effect on the saturation pressure of the particle gas phase.

(3) Since $t_e$ is much smaller than $t_{de}$, gas-phase heat exchange around the particle could be assumed to be quasi steady state. Since $t_d$ is much smaller than $t_{dr}$, it could be assumed that the temperature inside the particles is uniformly distributed.

(4) Since $t_{dr}$ and $t_h$ have the same order of magnitude, it could be assumed that the heating of the particles continues throughout the drying cycle and that the particle surface temperature is equal to the air temperature.

Based on the above assumptions, the following equations could be used to describe the drying of Baijiu yeast particles in spray-drying, and to predict the changes of moisture content and rehydration survival rate of yeast particles with drying time. Furthermore, the new sampling scheme could be used to sample and measure the spray-drying process of other microorganisms, to change the parameters of the mathematical model and extend it to the drying of all microorganisms.

The meanings and values of known or unknown variables used in the modeling process have shown in Table 2 below [20].

**Table 2.** Meanings and values of modeling parameters.

| Variable | Meaning | Value |
|---|---|---|
| $\rho_{wl}$ | Density of liquid water (25 °C) | 1000 kg m$^{-3}$ |
| $\rho_a$ | Density of water vapor (25 °C) | 1.2 kg m$^{-3}$ |
| $M_w$ | Relative molecular weight of water | $1.8 \times 10^{-3}$ kg mol$^{-1}$ |
| $M_a$ | Molecular weight of air | $2.8 \times 10^{-3}$ kg mol$^{-1}$ |
| $R_g$ | Universal gas constant | 8.314 J (mol K)$^{-1}$ |
| $P_{sat}^{ref}$ | Reference water saturation pressure | 4243 Pa |
| $T_{ref}$ | Reference temperature | 25 °C |
| $L_k$ | Latent heat of vaporization of water | 2436 Mj kg$^{-1}$ |
| $\delta_i$ | Diffusion coefficient of water vapor in air | $2.5 \times 10^{-5}$ m$^2$ s$^{-1}$ |
| $\nu$ | Kinematic viscosity of air | $1.7 \times 10^{-5}$ m$^2$ s$^{-1}$ |
| $\mu$ | Aerodynamic viscosity | $1.8 \times 10^{-5}$ Pa s |
| $c_{p,wl}$ | Specific heat capacity of liquid water | 4185 J (kg K)$^{-1}$ |
| $c_{p,a}$ | Specific heat capacity of air | 1000 J (kg K)$^{-1}$ |
| $\lambda_a$ | Air thermal conductivity | $2.62 \times 10^{-2}$ W (m K)$^{-1}$ |
| $D_0$ | Initial diameter of yeast spray particles | 100–400 μm (Determination by particle size) |
| $D$ | Diameter of yeast particles | 100–300 μm |
| $P_{tot}$ | Total pressure in drying tower | 101,325 Pa |
| $Y$ | Moisture content of air on a dry basis | 0.01 (d.b.) |
| $u$ | Hot wind speed | 6–8 m s$^{-1}$ |
| $X_0$ | Initial moisture content of yeast granules | 2.2–2.5 (d.b.) |
| $T_{g0}$ | Initial hot air temperature | 40–125 °C |
| $T_0$ | Initial yeast spray particle temperature | 28 °C |
| $t_{dr}$ | drying time | 14–22 s |
| $t_c$ | Contact time between gas flow and particles | $t_c \cong 2R\,/u \approx 0.001s$ |
| $t_{md}$ | Diffusion time of water vapor in particles | $t_{md} \cong R^2/\delta_1 \approx 0.01s$ |
| $t_{hd}$ | Thermal diffusion time inside yeast particles | $t_{hd} \cong R^2/k_{wl} \approx 2s$ |
| $t_h$ | Heating time of yeast particles in gas flow | $t_h \cong (1+X)C_{p,wl}/ah \approx 14-22s$ |
| $\tau$ | Tortuosity of dry porous shell | 4.45 |
| $T_g$ | surface temperature of yeast particles | Unknown |
| $\varepsilon$ | Porosity of yeast granules | Unknown |
| $k_d$ | Kinetic constant | Unknown |

*4.2. Mathematical Modeling*

4.2.1. Initial Conditions

The initial conditions were expressed as follows.

$$D(t=0) = D_0 \tag{6}$$

$$T_g(t=0) = T_{g0} \tag{7}$$

$$u_\infty = u_0 \tag{8}$$

$$X(t=0) = X_0 \tag{9}$$

$$X_e(t=0) = X_{e0} \tag{10}$$

$$X_d(t=0) = X_{d0} \tag{11}$$

In the Equations (6)–(11), $D$—diameter of yeast particles, mm, which could be measured by a laser particle sizer; $T$—yeast particle surface (hot air) temperature, °C; $u$—wind speed; $X$—water content, %; Each letter in the lower corner corresponds to each type of water, and 0 represents the initial value; $X_{d0}$—initial dry-basis content of bound water, %, determined by DSC scanning.

After conversion, Equation (6) corresponds to the spray flow (Particle size), Equation (7) corresponds to the hot air temperature, Equation (8) corresponds to the wind speed, and Equations (9)–(11) correspond to the moisture content at the transfer point. The initial conditions of the model were found by entering the equations one at a time.

### 4.2.2. Particle Size Modeling

According to the experimental results and pore theory, it could be considered that the diameter of yeast particles is a linear function of the water content of the particles, and it is isotropic. The real-time diameter D of the dried particles could be expressed as a function of the real-time moisture content $X_0$ with the coefficients $D_0$, $X_0$, and the shrinkage coefficient $\kappa$ as follows:

$$D = D_0\left(1 - \kappa\frac{X_0 - X}{X_0}\right) = \kappa D_0\frac{X}{X_0} + D_0 - \kappa D_0 \tag{12}$$

The shrinkage coefficient $\kappa$ of particles in Equation (12) could be estimated as $\kappa = 0.46$ based on a large number of experimental results in Figure 6.

### 4.2.3. Mass Transfer Modeling

The mass balance of water could be expressed as:

$$X = X_e + X_d + X_r \tag{13}$$

$$\frac{dX_e}{dt} = -J_e \tag{14}$$

$$\frac{dX_d}{dt} = -J_d \tag{15}$$

$$\frac{dX_r}{dt} = 0 \tag{16}$$

In Equations (13)–(16), $J_X$—Class $X$ water loss rate, where $X$ is representative of three categories of water, kg/s; $t$—time, s.

#### Class E Water

Class E water is in a pore channel network of pore throat diameter 0.1 μm made up of cytoplasm and cell membrane. Moreover, the dissipation rate $J_e$ of Class E water could be characterized by vapor diffusion in the particle dry zone (internal mass transfer coefficient $k_i$) and combined vapor diffusion convection (external mass transfer coefficient $k_e$) in the boundary layer around the particle as follows.

$$J_e = aM_w\frac{k_ek_i}{k_e + k_i}(C_{sat} - C_\infty) \tag{17}$$

In Equation (17), $a$—specific surface area, m$^2$/kg; $M_w$—relative molecular mass of water, kg/mol; $k_e$—external mass transfer coefficient of the particles; $k_i$—internal mass transfer coefficient of the particle; $C_{sat}$—the saturated concentration of water in air, mol/m$^3$; $C_\infty$—the concentration of water vapor in the air far from the particle, mol/m$^3$; As available by the ideal gas assumption, $C_{sat}$ and $C_\infty$ could be expressed as follows [21]:

$$C_{sat} = \frac{P_{sat}}{R_gT_g} \tag{18}$$

$$C_\infty = \frac{YM_wP_{tot}}{R_gT_g(M_a + YM_w)} \tag{19}$$

In Equations (18) and (19), $T_g$—yeast particle surface (hot air) temperature, °C, which could be measured by temperature sensor in the experiment; $T$—internal temperature of yeast particles, °C; $P_{tot}$—total pressure in drying tower, Pa; $P_{sat}$—water saturation pressure,

Pa. The relationship between $P_{sat}$ and the surface (hot air) temperature of yeast particles could be expressed by Clausius Clapeyron law as follows:

$$P_{sat} = P_{sat}^{ref} \exp(-\frac{L_k M_w}{R_g} \frac{T_{ref} - T_g}{T_g T_{ref}}) \tag{20}$$

The external mass transfer coefficient ($k_e$) in Equation (17) could be expressed as follows by the relation to the Sherwood's number ($sh$) of a fluid flowing around a particle [19]:

$$sh = \frac{k_e D}{\delta_i} = 2 + 0.6 \text{Re}^{0.5} Sc^{1/3} \tag{21}$$

In Equation (21), Reynolds number Re and Schmidt's number $Sc$ were defined as follows [22]:

$$\text{Re} = \frac{uD}{v} \tag{22}$$

$$Sc = \frac{v}{\delta_i} \tag{23}$$

The expressions for the external mass transfer coefficient ($k_e$) available by carrying Equations (22) and (23) into Equation (21) are as follows:

$$k_e = 2\delta_i + 0.6u^{1/2}\delta_i^{2/3}D^{-1/2}v^{-1/6} \tag{24}$$

The intra-particle mass transfer coefficient ($k_i$) in Equation (17), under the quasi steady state assumption, could be expressed in terms of the mass balance of water vapor inside the drying zone of the particles as follows:

$$\frac{d}{dt}(r\frac{dC}{dr}) = 0 \tag{25}$$

In Equation (25), the distance of the $r$—Class e water from the center of the particle, mm; $C$—concentration of water vapor in the voids of particles, mol/m$^3$; Its boundary conditions are:

$$C\Big|_{r=\frac{D}{2}} = C_{surf} \tag{26}$$

$$C\Big|_{r=\frac{D_h}{2}} = C_{sat} \tag{27}$$

In Equations (26) and (27): $C_{surf}$—concentration of water vapor in the contact air at the outer surface of the particle, mol/m$^3$; $C_{sat}$—the saturated concentration of water in the air, mol/m$^3$.

In Equation (25), $dC/dr$ could be expressed by the mass flux j away from the yeast particle water vapor as follows:

$$\frac{dC}{dr}\Big|_{r=\frac{1}{2}D_h} = -\frac{j\tau}{\varepsilon\delta_i} \tag{28}$$

Take Equations (26)–(28) into Equation (25):

$$-\frac{\varepsilon\delta_i}{\tau}\frac{dC}{dr}\Big|_{r=0.5D_h} = -\frac{\varepsilon\delta_i}{\tau}\frac{2}{D}\frac{1}{\ln^{(D)} - \ln^{(D_h)}}(C_{sat}\Big|_T - C_{surf}) = k_i(C_{sat}\Big|_T - C_{surf}) \tag{29}$$

The expression of internal mass transfer coefficient ($k_i$) could be obtained by sorting out Equation (29) as follows:

$$k_i = -\frac{\varepsilon\delta_i}{\tau}\frac{2}{D}\frac{1}{\ln\frac{D}{D_h}} \tag{30}$$

The moisture content and the diameter of the spheres could be considered to be directly proportional, assuming that all water is uniformly distributed and lost layer-by-layer within the spherical region where it was located (Figure 12), so the proportional relationship could be written as when starting drying (t = 0) and during drying (t = t):

$$\frac{4\pi}{3}\left(\frac{D}{2}\right)^3 = \alpha X_{e0} \tag{31}$$

$$\frac{4\pi}{3}\left(\frac{D_h}{2}\right)^3 = \alpha X_e \tag{32}$$

Dividing Equation (32) by Equation (31) was obtained:

$$\frac{D}{D_h} = \left(\frac{X_{e0}}{X_e}\right)^{1/3} \tag{33}$$

Class D Water

$J_d$ could be expressed by the first-order dynamic model [23],

$$J_d = k_d\left(X_d - X_{d0}\frac{X_e}{X_{e0}}\right) \tag{34}$$

In Equation (34), $k_d$—the kinetic constant related to the particle surface temperature $T_g$, s$^{-1}$.

In summary, the dispersion rates and the water content models for the two types of water available after finishing E and D are as follows:

$$J_e = \frac{-2\varepsilon\delta_i a M_w(2\delta_i + 0.6u^{1/2}\delta_i^{2/3}D^{-1/2}v^{-1/6})}{((2\delta_i + 0.6u^{1/2}\delta_i^{2/3}D^{-1/2}v^{-1/6})\tau D\ln\left(\left(\frac{X_{e0}}{X_e}\right)^{1/3}\right) - 2\varepsilon\delta_i}\frac{P_{sat}^{ref}\exp(-\frac{L_kM_w}{R_g}\frac{T_{ref}-T_g}{T_gT_{ref}})M_a + YM_w(P_{sat}^{ref}\exp(-\frac{L_kM_w}{R_g}\frac{T_{ref}-T_g}{T_gT_{ref}}) - P_{tot})}{R_gT_g(M_a + YM_w)} = \frac{dX_e}{dt} \tag{35}$$

$$J_d = k_d\left(X_d - X_{d0}\frac{X_e}{X_{e0}}\right) = k_dX_d = \frac{dX_d}{dt} \tag{36}$$

4.2.4. Heat Transfer Modeling

The macroscopic thermal equilibrium equation for yeast particles was as follows [24]:

$$ah(T_g - T) = (J_e + J_d)L_k + (1 + X)c_{p,wl}\frac{dT}{dt} \tag{37}$$

The relaxation time $t_h^*$ for the particle heating was defined as follows [25]:

$$t_h^* = \left(\frac{ah}{(1 + X)c_{p,wl}}\right)^{-1} \tag{38}$$

The expression of the heat transfer rate was obtained by bringing Equation (38) into Equation (37), as shown below:

$$\frac{dT}{dt} = \frac{1}{t_h^*}\left(T_g - T - \frac{(J_e + J_d)L_k}{ah}\right) \tag{39}$$

In Equations (37)–(39): $L_k$—vaporization latent heat of water, j/kg; $h$—heat exchange coefficient, W/km$^2$;

In Equations (38) and (39), the heat exchange coefficient $h$ between the particle and the surrounding air, could be expressed by the association formula of the Nussle number (Nu) of the fluid flow around the particle as follows [26]:

$$Nu = \frac{hD}{\lambda_a} = 0.3 + \frac{0.62\text{Re}^{0.5}{P_r}^{1/3}}{\left(1 + \left(\frac{0.4}{\text{Pr}}\right)^{2/3}\right)^{1/4}} \tag{40}$$

In Equation (40): Re—Reynolds number; Pr—Prandtl number. The definition formula is:

$$\text{Pr} = \frac{\mu c_{p,a}}{\lambda_a} \tag{41}$$

The heat transfer rate calculation model could be obtained by successively bringing Equations (38)–(41) into Equation (37):

$$\frac{dT}{dt} = \left[\frac{0.3a\lambda_a}{(1+X)c_{p,wl}} + \frac{0.62a\text{Re}^{0.5}\mu^{1/3}c_{p,a}^{1/3}\lambda_a^{2/3}}{D\left(1+\left(\frac{0.4\lambda_a}{\mu c_{p,a}}\right)^{2/3}\right)^{1/4}(1+X)c_{p,wl}}\right]\left[T_g - T - \frac{(J_e + J_d)L_k D\left(1+\left(\frac{0.4\lambda_a}{\mu c_{p,a}}\right)^{2/3}\right)^{1/4}}{0.3a\lambda_a D\left(1+\left(\frac{0.4\lambda_a}{\mu c_{p,a}}\right)^{2/3}\right)^{1/4} + 0.62a\text{Re}^{0.5}\mu^{1/3}c_{p,a}^{1/3}\lambda_a^{2/3}}\right] \tag{42}$$

Since the water in yeast particles was divided into two types, E and D, which are lost in turn, Equation (42) should be divided into two parts as follows:

$$\frac{dT}{dt} = \left[\frac{0.3a\lambda_a}{(1+X_e)c_{p,wl}} + \frac{0.62a\text{Re}^{0.5}\mu^{1/3}c_{p,a}^{1/3}\lambda_a^{2/3}}{D\left(1+\left(\frac{0.4\lambda_a}{\mu c_{p,a}}\right)^{2/3}\right)^{1/4}(1+X_e)c_{p,wl}}\right]\left[T_g - T - \frac{J_e L_k D\left(1+\left(\frac{0.4\lambda_a}{\mu c_{p,a}}\right)^{2/3}\right)^{1/4}}{0.3a\lambda_a D\left(1+\left(\frac{0.4\lambda_a}{\mu c_{p,a}}\right)^{2/3}\right)^{1/4} + 0.62a\text{Re}^{0.5}\mu^{1/3}c_{p,a}^{1/3}\lambda_a^{2/3}}\right] \tag{43}$$

$$\frac{dT}{dt} = \left[\frac{0.3a\lambda_a}{(1+X_d)c_{p,wl}} + \frac{0.62a\text{Re}^{0.5}\mu^{1/3}c_{p,a}^{1/3}\lambda_a^{2/3}}{D\left(1+\left(\frac{0.4\lambda_a}{\mu c_{p,a}}\right)^{2/3}\right)^{1/4}(1+X_d)c_{p,wl}}\right]\left[T_g - T - \frac{J_d L_k D\left(1+\left(\frac{0.4\lambda_a}{\mu c_{p,a}}\right)^{2/3}\right)^{1/4}}{0.3a\lambda_a D\left(1+\left(\frac{0.4\lambda_a}{\mu c_{p,a}}\right)^{2/3}\right)^{1/4} + 0.62a\text{Re}^{0.5}\mu^{1/3}c_{p,a}^{1/3}\lambda_a^{2/3}}\right] \tag{44}$$

### 4.2.5. Rehydration Survival Rate Modeling

Thermal inactivation of microbial cells is usually expressed using the first-order kinetic equation as follows [27]:

$$\frac{dQ_v}{dt} = -k_r Q_v \tag{45}$$

In Equation (45), $k_r$—deactivation rate constant. Since dehydration damage and thermal damage are the main factors causing yeast inactivation in spray drying, available from the above heat transfer modeling, the change of temperature is also closely related to the moisture content within the yeast particles. So in this study, it was assumed that thermal inactivation is the most dominant inactivation way of yeast, and $k_r$ could be correlated with the internal temperature $T$ of the particles using the Arrhenius equation as follows [28]:

$$k_r = k_0 e^{\frac{-E_d}{RT}} \tag{46}$$

In Equation (46), $k_0$—pre exponential factor, $E_d$—activation energy of inactivated cells, $R$—gas constant.

Carrying Equation (46) into Equation (45) was available:

$$\frac{dQ_v}{dt} = -Q_v k_0 e^{\frac{-E_d}{RT}} \tag{47}$$

### 4.3. Model Description

The initial parameters of spray drying were input according to Equations (4)–(9), and then the values of the value parameters in Table 2 were brought into Equations (10), (33), (34), (41), (42) and (45). Then, according to the mathematical model operation logic shown in Figure 13, the function curve of moisture content and rehydration survival rate of dried particles in spray drying process with time could be obtained.

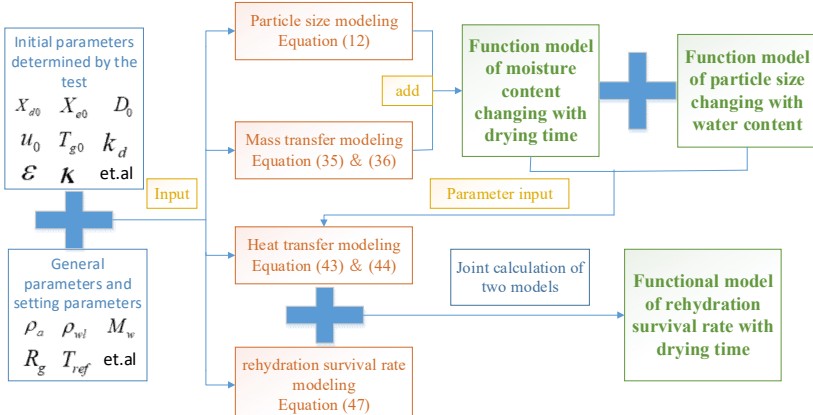

**Figure 13.** Liquor yeast diagram of mathematical model operation logic.

### 4.4. Determination of Model Parameters

As shown in Table 2, there are also three modeling parameters $\varepsilon$, $X_{d0}$, and $k_d$ that will change with the change of the spray-drying process and microbial species. Therefore, this study plans to take Baijiu Yeast as an example to determine the changing model parameters, to guide the determination of other microbial spray-drying modeling parameters.

$\varepsilon$ is the porosity before the drying of yeast pellets, which was determined by a large number of trials in a modelling study by debase on the hot-air drying of yeast pellets $\varepsilon = 0.58$ [18].

As shown in Figure 12, it was assumed that the three types of water E, D, and r are uniformly distributed in the circle from the outside to the inside in that order, and are lost in turn. Based on the ease of water dispersal and the degree of damage to yeast in each category, $X_{e0}$ = 1.8–2.2 (d.b), $X_{d0}$ = 0.3–0.4 (d.b), and $X_r$ = 0.05–0.11 (d.b) could be tentatively determined, which is the yeast particles dry-based water content at the water conversion point of Class E and Class D and was about 0.3–0.4 (d.b) [18].

The moisture content (moisture state) of the bound water on a dry basis, also known as the moisture content at the transition point, could be accurately measured by differential scanning calorimeter (DSC) in the experiment and used as the input parameter of the model. The value of bound water dry-basis moisture content $X_{d0}$ of yeast dry granules obtained under various spray-drying parameters is shown in Table 3 below, and $X_{d0}$ = 0.35 (d.b) was taken after comparative analysis.

**Table 3.** Dry-basis content of bound water (d.b).

| Inlet Hot Air Temperature (°C) | | 95 | | 105 | | 115 | | 125 | |
|---|---|---|---|---|---|---|---|---|---|
| Hot air inflow | 60 | $0.35 \pm 0.012$ | $0.36 \pm 0.01$ | $0.34 \pm 0.017$ | $0.35 \pm 0.009$ | $0.35 \pm 0.013$ | $0.35 \pm 0.010$ | $0.36 \pm 0.004$ | $0.34 \pm 0.010$ |
| (m³/h) | 70 | $0.35 \pm 0.004$ | $0.36 \pm 0.011$ | $0.35 \pm 0.012$ | $0.34 \pm 0.014$ | $0.36 \pm 0.008$ | $0.34 \pm 0.017$ | $0.35 \pm 0.011$ | $0.36 \pm 0.007$ |
| Material inflow (mL/h) | | 600 | 700 | 600 | 700 | 600 | 700 | 600 | 700 |

Because b < 0.4 (D.B), $k_d$ could be determined by linear regression $\ln(X - X_r)$, and there is $X = X_d + X_r$, that is, the change of $X_d$ with time could be expressed as follows by Equation (34) [29]:

$$X_d = X_{d0} \exp(-k_d t) \tag{48}$$

Take the logarithm of Equation (48) to obtain:

$$k_d = -\frac{\ln(X_d) - \ln(X_{d0})}{t} = -\frac{\ln(X - X_r) - \ln(X_{d0})}{t} \tag{49}$$

## 5. Results and Analysis

Firstly, based on four kinds of inlet hot air temperatures, a group of typical process parameters were selected for spray-drying of Baijiu Yeast. Then, the test data were brought into the above established mathematical model to obtain the corresponding prediction function, then Origin was used to make a diagram. Finally, the predicted function curve of the mathematical model of the Particle size, moisture content, and rehydration survival rate of yeast particles with time has been compared with the experimental results, as shown in Figures 14–16 respectively.

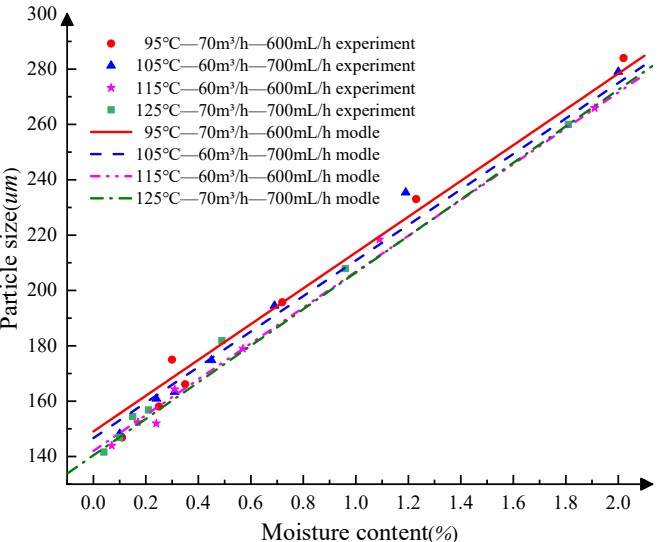

**Figure 14.** Comparison between model curve and experimental value of yeast particle size changing with moisture content.

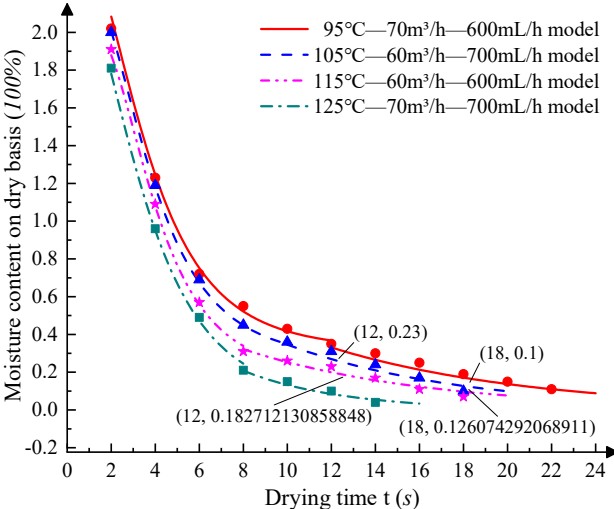

**Figure 15.** Model curve of yeast particle dry based moisture content as a function of time contrasting with experimental values.

As shown in Figure 14, the coincidence rate of the function curve of dry particle size predicted by the mathematical model with its dry-basis moisture content and the experimental data is as high as 96%, which proves that the mathematical model could better predict the historical change of dry particles during spray drying.

According to the analysis in Figures 15 and 16, the prediction curves and experimental values of the heat and mass transfer mathematical model of porous media based on real-time sampling data for the moisture content and rehydration survival rate of yeast particles in the spray drying process were obtained when the inlet temperature was between 95 °C

and 125 °C. When the inlet hot air temperature is 115 °C—the hot air flow is 60 m$^3$/h—the material flow is 600 mL/h, the maximum errors $X_{\max} = 0.03(d.b)$ and $Q_{v\max} = 6.43\%$ were taken when the sampling time was 12 s. In the dry DC circulation area, the moisture content and rehydration survival rates predicted by the model almost coincide with the experimental values. After drying, the predicted water content and rehydration survival rate were lower than the experimental values. However, they were close to overlapping at the end.

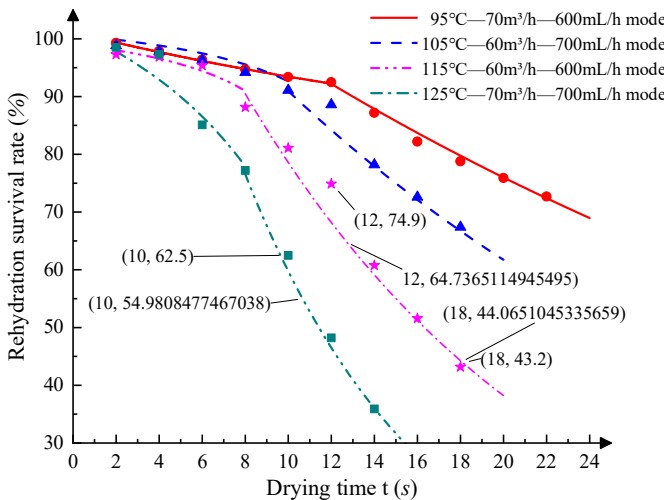

**Figure 16.** Comparison between model curve and experimental value of yeast granule rehydration survival rate with time.

The predicted and experimental values of the water bearing rate and rehydration survival rate of yeast particles during the drying process show that the predicted values in the DC circulation area almost coincide with the end point, while the predicted values in the recirculation circulation area are less than the experimental values. The main reason is that in the reflux circulation area, the sampling cup collects both reflux dry particles and falling dry particles, resulting in the moisture content and rehydration survival rate of dry particles measured in the test being higher than the predicted value. Apart from the recirculation area, the maximum error values of the model for predicting the moisture content and rehydration survival rate of yeast dry particles during spray-drying are $X_{\max 1} = 0.027(d.b)$ and $Q_{v\max 1} = 1.06\%$, respectively, which are less than 5%. It was proven that the mathematical model of heat and mass transfer in porous media-based on real-time sampling data could accurately predict the moisture content and rehydration survival rate of microbial particles in the process of spray-drying, and provide guidance for revealing the changes of microbial particles in the process of spray-drying. At the same time, it was proven again that the inference of particle trajectory in the process of spray drying was correct.

## 6. Conclusions

In this study, a sampling system for real-time sampling, rehydration, and non-destructive storage of microbial dry particles in the spray-drying process (which has been submitted for a Chinese invention patent) was introduced. With the cooperation of later detection, the system could reveal the changes of microorganisms and their particles in the high-speed, complex, and invisible spray-drying process, which lays a foundation for later mathematical modeling to predict and explore the microscopic damage mechanism of microorganisms in the spray drying process. Based on the measured data, the mathematical model of Baijiu yeast granule drying was established according to the law of heat and mass transfer and the theory of porous media. Comparing the predicted value of the mathematical model with the experimental results, the maximum error values of moisture content and rehydration survival rate are only $X_{\max 1} = 0.027(d.b)$ and $Q_{v\max 1} = 1.06\%$, respectively, both are less than 5%, which proves

that the model could accurately predict the changes of moisture content and rehydration survival rate of microbial dried particles with drying time in the process of spray-drying.

Based on the analysis of the experimental data and the predicted values of the model, it was concluded that the spray-drying process could be divided into DC circulation area and reflux circulation area. In the DC circulation area, the yeast particles have enough kinetic energy to overcome the resistance of the laterally flowing air in the vortex area, move along the Original spray path, and dehydrate rapidly. In the recirculation area, the kinetic energy of yeast particles was weakened, which is not enough to overcome the resistance of the lateral air flow in the vortex area, but was pulled up along the wall direction and further reduced for dehydration. The moisture content $X_z$ = 0.3–0.4 (d.b) of yeast particles transferred from the DC circulation area to the reflux area. After entering the reflux circulation area, the yeast begins to inactivate rapidly.

Due to the lack of yeast activity detection equipment, it is impossible to quantitatively detect the activity of yeast in dry granules. Therefore, it is impossible to conduct mathematical modeling and prediction on the change of yeast activity with time in the drying process and further verify the correctness of the mathematical model. In follow-up studies, the author will seek better experimental conditions and conduct in-depth research.

**Author Contributions:** Conception and writing, F.-k.X.; Review and editing, Y.-j.Y.; Language polish, Y.-y.X.; Supervision and data curation, J.-y.L.; Review, Z.Z.; Language polish, L.-b.T. All authors have read and agreed to the published version of the manuscript.

**Funding:** This work was supported by the National Natural Science Foundation of China [grant number 51876109]; the National Key R&D Program of China [grant number 2017YFD0400902-1]; and the Key Project of International Science and Technology Cooperation Program for Shaanxi Province [grant number 2020KWZ-015]; Scientific research plan project of youth innovation team of Shaanxi Provincial Department of Education [grant number 22JP012].

**Institutional Review Board Statement:** Not applicable.

**Informed Consent Statement:** Not applicable.

**Data Availability Statement:** Not applicable.

**Conflicts of Interest:** This is a research article, researched by the author himself, without conflict of interest with any individual or collective, hereby declares.

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
