# Peer review of "Modeling Study of a Microbial Spray-Drying Process Based on Real-Time Sampling"

_processes, doi:10.3390/pr10091789_

Round 1

Reviewer 1 Report

My comments are very general, I hope they are useful to you.

1. Some numbers of the references to the text are as a flown number and others are not.

2. In figure 4, I don't find number 26 or number 39 on the schematic.

3. It is not understood what they want to explain with table 1.

4. The mathematical model is extensively described and figure 12 facilitates the reading of the model and its understanding.

5. It would be important to include some limitations that they have had for the development of the work and what remains to be done.

6. Check that the references are written in the estimate requested in the indications for auters (homogenize).

Author Response

Thank you for your letter and for the Reviewers’ comments concerning our submitted manuscript entitled “Modeling study of microbial spray-drying process based on real time sampling system” (ID: processes-1878817). Those comments are all valuable and very helpful for revising and improving our paper, as well as the important guiding significance to our researches. We have studied comments carefully and have made correction which we hope meet with approval. After the language issue editing by our colleague Ying Han, revised portion are marked in red and Blue shading in the revised manuscript. The response details are shown in the “Response to Reviewers comments”.

Reviewer 2 Report

Very complete paper but from the editing point of view unacceptable. Extensive editing has to be applied to the text - for example titles of chapters once start from a capital letter and once from a small letter. Some captures under figures are typed as a lower index. Some calling of equations is wrong - Eq. 45 is not a logarithmic form of Eq. 30. And so on and so on. As the reviewer, I will appreciate seeing the improved version of the paper.

Author Response

Thank you for your letter and for the Reviewers’ comments concerning our submitted manuscript entitled “Modeling study of microbial spray-drying process based on real time sampling system” (ID: processes-1878817). Those comments are all valuable and very helpful for revising and improving our paper, as well as the important guiding significance to our researches. We have studied comments carefully and have made correction which we hope meet with approval. After the language issue editing by our colleague Ying Han, revised portion are marked in red and Blue shading in the revised manuscript. The response details are shown in the “Response to Reviewers comments”.

Response to reviewer comments:

Point 1: Very complete paper but from the editing point of view unacceptable. Extensive editing has to be applied to the text - for example titles of chapters once start from a capital letter and once from a small letter. Some captures under figures are typed as a lower index. Some calling of equations is wrong - Eq. 45 is not a logarithmic form of Eq. 30. And so on and so on. As the reviewer, I will appreciate seeing the improved version of the paper.

Response1: Thank you for your comment. According to your suggestions, we have carefully checked the formula and reference number of the full text, and modified the case format of the first letter of the chapter title. At the same time, an English professor was hired to polish the English expression of the full text, and the part modified after the language polishing was marked with blue background.

First of all, the call of formula is checked and it is found that: ① In line 531, equation 47 is obtained by taking the logarithm of equation 46 instead of taking the logarithm of equation 30. ② There is no explanation for the variables in EQ 1 in the original text, so lines 107 and 108 are inserted to explain the variables in EQ 1. ③ The original text reverses the position of formula 36 and formula 37, and the correct expression should be: “bring formula 36 into formula 35 and get the expression of heat transfer rate, as shown below.” It has been modified in lines 464 to 468 in the pepper. ④ The description of equation 40 was omitted in the original writing, so the description of equation 40 was inserted into lines 476 and 477. And so on and so on.

Then, the initials of chapter titles of the full text were carefully checked, and the initials of all titles were changed to uppercase format as follows: 2.1. Materials and methods. 2.2. Test procedure. 2.2.2. Spray-drying test design. 3.1. Common data values and assumptions. 3.2.1. Initial conditions. 3.2.2 Particle size modeling. 3.2.4. Heat transfer modeling. 3.3. Model description. 3.4. Determination of model parameters. 4. Results and analysis. 5. Conclusion.

And then, we have replaced references [1], [4], [15] and [23] which have little relevance to the content. Then we have reviewed the references again to confirm that they are relevant to this article.

[1] Tay, J.B.J.; Chua, X.; Ang, C.; Subramanian, G.S.; Tan, S.Y.; Lin, E.M.J.; Wu, W.-Y.; Goh, K.K.T.; Lim, K. Effects of Spray-Drying Inlet Temperature on the Production of High-Quality Native Rice Starch. Processes 2021, 9, 1557. https://doi.org/10.3390/pr9091557  

[4] Al Zaitone, B.; Al-Zahrani, A.; Ahmed, O.; Saeed, U.; Taimoor, A.A. Spray Drying of PEG6000 Suspension: Reaction Engineering Approach (REA) Modeling of Single Droplet Drying Kinetics. Processes 2022, 10, 1365. https://doi.org/10.3390/pr10071365

[15] Bushnaq, H.; Krishnamoorthy, R.; Abu-Zahra, M.; Hasan, S.W.; Taher, H.; Alomar, S.Y.; Ahmad, N.; Banat, F. Supercritical Technology-Based Date Sugar Powder Production: Process Modeling and Simulation. Processes 2022, 10, 257. https://doi.org/10.3390/pr10020257  

[23] Koumbogle, K.; Gitzhofer, F.; Abatzoglou, N. Moisture Transport Coefficients Determination on a Model Pharmaceutical Tablet. Processes 2022, 10, 254. https://doi.org/10.3390/pr10020254

Finally, Professor Liang Chun, an English major, was invited to embellish the paper. In order not to conflict with the red marking of the revised part of the content, the revised part after the language embellishment was marked in the original text with blue shading. The specific language is modified as follows:

Line 9-Line 25: Abstract: The process of microbial spray-drying has inherent defects such as short time, complex and non-visualization of particle trajectory. However, there has been a lack of effective methods for real-time sampling, rehydration and non-destructive storage of dried particles, as well as mathematical modeling of the drying process of yeast particles based on sampling and measurement data.  Therefore, the author firstly developed a real-time sampling system of spray-dried particles, which completed real-time sampling, rehydration and non-destructive storage of spray-dried particles, and realized the real reproduction of the changes of yeast particles in the process of spray-drying. The laws that the motion trajectory of microbial particles during spray-drying has divided into the first cycle region and the reflux cycle region are concluded, and the partition mechanism is explored. Then, based on the sampling data and the law of heat and mass transfer, a mathematical model of porous media was established to predict the variation of moisture content and rehydration survival rate of dried microbial particles with drying time. Finally, the mathematical model was tested by microbial spray drying experiment, and the maximum error between the predicted value and the test value of moisture content and rehydration survival rate was  andrespectively, both less than 5%,which proved the correctness of the mathematical model of porous media and laid a foundation for the study of the damage mechanism of microbial spray drying.

Line 29-Line 105: 0 Introduction:

Microbial desiccation is the core research topic of livelihood fields (such as food, medicine, etc.). As the most widely used microbial drying process, spray drying has the advantages of high efficiency, low cost and continuous production. It not only occupies a dominant position in the market, but also is a hot spot in current research. Nevertheless, due to the inherent defects of the microbial spray drying process, such as short time, complex and invisible trajectory of spray particles, it is very difficult to directly sample and study the drying process of microbial particles. Therefore, there has been a lack of effective means for real-time sampling, rehydration and non-destructive storage of dried particles, and further establishing mathematical models to accurately predict the moisture content and rehydration survival rate of microbial particles in the spray drying process. At present, the common research method is " single droplet simulation experiment + mathematical modeling " On the one hand, the single droplet simulation experiment could not reproduce the environment such as short residence time, complex movement trajectory and high-speed collision of spray-dried particles, and the experimental data could not truly reflect the changes of microorganisms in the  spray-drying process. On the other hand, most mathematical models ignore changes in the trajectory of a dry particle and simplify the trajectory to top-down homogeneous motion. Therefore, the mathematical models established by this combination method can not predict the spray-drying process of microorganisms well.

Dr. Gong Pimin (Harbin Institute of Technology), proposed a vertical arrangement of single-cup straight cylinder sampling system. Although this system can complete the real-time sampling of spray-dried particles, it suffers from the disadvantages of poor sampling stability and nondestructive storage, which could not reproduce the real microbial changes during spray-drying. Previous researches on modeling mainly focus on regression analysis to build empirical models, such as Akpinar equal to the thin layer model proposed in 2006, there is little in-depth exploration of heat and mass transfer and microbial cell damage of drying process, so the prediction accuracy is not high. In 2013, L. Spreutels, a Canadian scholar, established an image-only mathematical model based on the dynamic drying curve method based on the single droplet simulation experiment, although the spray particles were assumed to fall at a uniform speed in the spray drying process. However, the model was based on the single drop test, so the rehydration survival of yeast particles during drying could not be accurately predicted as a function of drying time.

In response to the above problems, the author firstly developed a novel microbial spray-drying real-time sampling system (applied to China invention patent) which can implement real-time sampling, rehydration and nondestructive storage on spray dried particles. Then, on the basis of real-time sampling data, the heat and mass transfer model of porous media proposed by the research team in the previous study was used to complete the mathematical modeling of spray drying process. Finally, through the comparison and analysis with the experimental data, it is proved that the model can accurately predict the moisture content and rehydration survival rate of dried particles with time in the process of microbial spray drying.

  1. A brief introduction of the new sampling system

1.1.A brief introduction of the sampling arrangement of double wet and dry samples in step - ladder type

In the novel sampling scheme, a dry pellet sampling cup and a rehydration sample sampling cup, 50 mm apart, are arranged on the same ladder step and are responsible for collecting the dry pellets and real-time rehydration of the dry pellets, respectively, with nondestructive storage. The drying tower is then bisected equally distantly by height and perimeter into 6 elevations and 7-segment arcs, respectively, with each ladder step again arranged at the intermediate value of height and radians, completing the 7-Stage sampling arrangement. In order to prevent the spray particles from directly entering the sampling cup and affecting the sampling stability, arc-shaped baffles were installed at 30mm above the mouth of each sampling cup. Finally, the design of stepped-type dry and wet double sample sampling arrangement was completed. This sampling scheme, with a step ladder type design, avoids interference between individual sampling cups in a vertical type arrangement scheme, which, combined with the installation of arcuated baffles, greatly improves the stability and accuracy of sampling.

1.2. A brief review of two cup surface sampling cups

In terms of sampling principle, the sampling cup attracts dry particles loaded with microorganisms and carboxyl iron powder (CIP) into the sampling cup through the magnetic field of a strong magnet at the bottom of the outer cup, so as to achieve the purpose of stable sampling.

At constant temperature and humidity, the sampling cup is filled with -40 ℃ antifreeze for automobile in the freezing layer, and then put into the -60 ℃ refrigerator for quick freezing to form a solid state, so as to maintain the low temperature environment in the inner cup. At the same time, the thermal and magnetic exchange between inside and outside of the sampling cup is isolated by pasting the thermal insulation magnetic patch evenly on the outer surface of the outer cup, and the low temperature environment in the inner cup is further maintained, while the strong magnetic field is prevented from penetrating the outer cup and affecting the sampling process.

In order to improve the sampling stability and constant temperature water holding capacity at the same time, the sampling cup is improved to design a straight cylinder inner cup to be a convex curved inner cup by rotating the first 20mm of the Bezier cubic curve with control points , ,  and ,curve group is shown in Fig. 1 below, and the equation of curve group is shown in eq1 below.

Line111-Line 168: The curved inner cup structure changes the air flow field, wind speed and pressure field in the cup and reduces the inflow of external hot air (as shown in Fig. 2). Finally, the purpose of improving the constant temperature water holding capacity of the sampling cup without increasing the depth of the sampling cup to ensure the sampling capacity and stability is achieved.

In order to achieve non-destructive installation, the sampling cup is fixed by connecting magnet and spray drying tower to achieve the purpose of non-destructive installation. In addition, the four exposed surfaces of the magnet were affixed with thermal insulation magnetic patches to prevent them from affecting the spray-drying sampling process.

In order to measure temperature and humidity in real time, the sampling cup records the sampling temperature and humidity at the sampling point in real time through a pasted wireless temperature and humidity sensor placed in the outer cup mouth, and draws the curve of temperature and humidity change.

The overall structure diagram of the sampling cup is shown in Fig.3 below. The size of the sampling cup is: the upper surface of the inner cup is 20mm high, the minimum inner diameter is 18mm; The lower part of the inner cup is 10mm high and the inner diameter is 24mm; The height of the outer cup is 65mm, the outer diameter is 42mm; The wall thickness is 2mm.

A large number of experiments have proved that the combination of the double cup curved sampling cup and step ladder type sampling scheme can achieve stable real-time sampling, rehydration and non-destructive storage of spray drying particles, and the sampling data can accurately reflect the changes of microorganisms in the process of spray drying.

  1. Experimental design and data analysis

In this test, Baijiu yeast was sampled by a novel sampling system using three parameters: spray-dried material flow, hot air wind speed and temperature as test variables, and the temperature and humidity at each sampling site were recorded. Then, the dry based water content, moisture status and particle size of the dried pellets in each dried pellet sampling cup were determined, respectively, and the rehydration survival rate of yeasts in each rehydrated sample sampling cup was determined. Finally, the determination parameters were analyzed to obtain the movement law of microbial particles during spray-drying.

2.1. Materials and methods

The schematic diagram of the spray-drying test device is shown in Fig. 4. The left figure is the overall structure diagram, and the right figure is the structure diagram of sampling and temperature and humidity detection in the drying tower. The parameters of the spray-drying tower used in the experiment are tower height h =900mm, cylinder diameter d=400mm, conical taper at the bottom a=60 °, and tower bottom diameter d=150mm.

As shown in Fig. 4 above, the temperature and humidity measurement of the sampling point is completed by the temperature and humidity sensors arranged at the mouth of the sampling cup in cooperation with the data acquisition system. The dry base moisture content, water state, particle size and rehydration survival rate of samples in the sampling cup were determined by water meter, differential scanning calorimeter (DSC), Winner3001 dry powder laser particle size meter and hemocytometry, respectively.

Line176-Line 181: According to the research team prior studies:①The optimized N medium formulation were yeast extract 15 g/L, peptone 15 g/L, beef extract 15 g/L, fructose 40 g/L, KH2PO4 3 g/L, and inositol 0.1 g/L. ②Yeast desiccant protectant formulations were sucrose 100 g/L, trehalose 100 g/L, lactose 120 g/L, and sorbitol 80 g/L. ③The optimal vehicles when dried were reconstituted skimmed milk powder (RSM) at a concentration of 30% in culture broth. ④ Carboxyiron powder (CIP) was added as follows: CIP: RSM = 1:8.

Line254-Line 282: As shown in Fig. 5 above, with increasing sampling height, the measured sampling point temperatures at the cup mouth of the sampling cup, all showed a decreasing trend of speed reduction. The rate of temperature drops at sampling site decreased obviously after sampling height of 15 cm and stabilized near the outlet temperature at sampling site temperature when sampling height of 45 cm.

As shown in Fig. 8 above, the moisture content of dry particles decreases with the increase of drying time at a constant rate first and then at a reduced rate.

As shown in Fig. 7, when the sampling height was less than 45 cm, the dry based water content of yeasts all decreased rapidly with increasing sampling height. After sampling height was greater than 45 cm, under each drying parameter, with the increase of sampling height, the dry basis water content of yeast, however, showed different variation rules as follows. ① In curve 2 of group c and all curves of group d, the dry basis moisture content of yeast decreased with the increase of sampling height. ② In all the curves except curve. 2 In group B and C, the moisture content of dry basis at the sampling height of 75cm is lower than that at 90cm. ③ In all curves of group A, the moisture content of dry basis is lower than 75cm at the sampling height of 60cm and then lower than 90cm.

Based on the analysis of Fig. 5 and Fig. 7, it could be concluded that in the constant rate drying stage, yeast particles are rapidly dehydrated in a high-temperature environment, which makes the temperature in this stage decrease rapidly, and the sampling height is 45cm, which is close to the outlet temperature of spray-drying. In the subsequent stage, yeast particles are dried at a low rate in a medium and low temperature environment close to the outlet temperature.

Combined with the analysis of fig 7 and fig 8, yeast particles are not entirely top-down moving. Firstly, in group-c curve 2 and all curves in group-d, yeast particles move from top to bottom without reflux. Secondly, in all the curves except curve 2 in group B and group C, yeast particles refluxed from the sampling height of 90cm to 75cm, and then decreased along the tower wall. Finally, in all curves of group A, the yeast particles refluxed from the sampling height of 90cm to 60cm, and then decreased along the tower wall. The motion track is shown in Fig. 9.

Line308-Line 312: To establish the mathematical prediction model based on the experimental parameters, reasonable assumptions must be made in combination with the experimental data, and then the physical model of the spray drying process and yeast particles should be simplified. Therefore, based on the experimental results, this study makes five hypotheses as follows:

Line324-Line 333: Since cell membrane is the most typical semi-permeable membrane, which has no effect on the entry and exit of small molecules such as water molecules, in this study, the water in the outer protective agent of cell membrane and the flowing water that cells do not combine with cell structures are uniformly classified as class E free water. According to the pore network theory, it is assumed that the pore is uniformly distributed in a microscopic pore network (pore throat) with a diameter of 0.1um, and its loss is mainly obstructed by capillary forces in porous media and saturated vapor pressure on particle surface. The presence of class E water maintained the gas-liquid two-phase balance on the surface of dried particles, and kept the bacteria in the "wet bulb temperature" at the early stage of drying, and the yeast was almost not damaged by heat (survival rate >95%).

Line324-Line 333: (2)Since the radius of the grid channel of the protective agent outside the yeast body is 10, the capillary effect has negligible effect on the saturation pressure of the particle gas phase.

Line503-Line 505: Then, according to the mathematical model operation logic shown in Fig.13, the function curve of moisture content and rehydration survival rate of dried particles in spray drying process with time can be obtained.

Line554-Line 563: According to the analysis in Fig. 15 and 16, the prediction curves and experimental values of the heat and mass transfer mathematical model of porous media based on real-time sampling data for the moisture content and rehydration survival rate of yeast particles in the spray drying process were obtained when the inlet temperature was between 95 ℃and 125 ℃When the inlet hot air temperature is 115 ℃ - the hot air flow is 60 - the material flow is 600, the maximum errors  and are taken when the sampling time is 12s. In the dry DC circulation area, the moisture content and rehydration survival rate predicted by the model almost coincide with the experimental values. After drying, the predicted water content and rehydration survival rate were lower than the experimental values. But it's close to overlap at the end.

Line580-Line 586: In this study, a sampling system for real-time sampling, rehydration and non-destructive storage of microbial dry particles in spray drying process (which has been applied for a Chinese invention patent) is introduced. With the cooperation of later detection, the system can reveal the changes of microorganisms and their particles in the high-speed, complex and invisible spray drying process, which lays a foundation for later mathematical modeling to predict and explore the microscopic damage mechanism of microorganisms in the spray drying process.

Line594-Line 601: Based on the analysis of the experimental data and the predicted values of the model, it is concluded that the spray-drying process can be divided into DC circulation area and reflux circulation area. In the DC circulation area, the yeast particles have high enough kinetic energy to overcome the resistance of the laterally flowing air in the vortex area, move along the original spray path and dehydrate rapidly. In the recirculation area, the kinetic energy of yeast particles is weakened, which is not enough to overcome the resistance of the lateral air flow in the vortex area, but is pulled up along the wall direction and further reduced for dehydration.

This paper represented our original work. It had not been submitted to any other journal for publication. We all strongly believed that our research paper may be particular interest to the readers of your journal. Thank you very much for your consideration of our revised manuscript for potential publication, and we would be happy to receive your feedback as soon as possible and revise it if required.

Best regards,

Sincerely,

Fengkui Xiong Doctor

29 August 2022

Reviewer 3 Report

* Please, in the line 32 mention the advantages of the spray drying technique.

* In the line 33 clarify why short time is an inherent defects, since it is an advantage at the industrial level.

* In lines (197 - 198) should be described the methodology what was used to determine the moisture content, moisture state, and particle size. That is, how the moisture content is determined by moisture meter, the moisture state by DSC and particle size by laser granulometry.

* The results of moisture state by DSC and particle size by laser granulometry are not shown in the article.

* A comparison should be made in the moisture content, moisture state, and particle size of the traditional spray drying procedure and the model proposed in this work.

Author Response

(The authors gave the same response as above.)

Round 2

Reviewer 2 Report

A lot of the editorial work has been done. So I think that the paper should be accepted for the publication.

Author Response

Author's thanks to Reviewers

Dear Reviewer:

Thank very much for your kind work and consideration on publication of our paper. On behalf of my co-authors, we would like to express our great appreciation to editor and reviewers, Thank you and best regards. Your sincerely, Feng-kui XIONG.

Corrisponding author: name: Yue-jin YUAN.

e-mail: [email protected].

Fengkui Xiong Doctor

01 September 2022

Reviewer 3 Report

The authors made all the necessary changes to improve the manuscript, and now I recommend it for publication in its current form. 

Author Response

(The authors gave the same response as above.)
